# Eucalypt Extracts Prepared by a No-Waste Method and Their 3D-Printed Dosage Forms Show Antimicrobial and Anti-Inflammatory Activity

**DOI:** 10.3390/plants13060754

**Published:** 2024-03-07

**Authors:** Oleh Koshovyi, Mykola Komisarenko, Tatyana Osolodchenko, Andrey Komissarenko, Reet Mändar, Siiri Kõljalg, Jyrki Heinämäki, Ain Raal

**Affiliations:** 1Institute of Pharmacy, Faculty of Medicine, University of Tartu, Nooruse 1, 50411 Tartu, Estonia; oleh.koshovyi@ut.ee (O.K.); jyrki.heinamaki@ut.ee (J.H.); 2Pharmacognosy Department, The National University of Pharmacy, 53 Pushkinska St., 61002 Kharkiv, Ukraine; gnosy@nuph.edu.ua (M.K.); a0503012358@gmail.com (A.K.); 3State Institution “I.Mechnikov Institute of Microbiology and Immunology, National Academy of Medical Sciences of Ukraine”, 14-16, Pushkinskaya St., 61057 Kharkov, Ukraine; imi_lbb@ukr.net; 4Department of Microbiology, Institute of Biomedicine and Translational Medicine, Faculty of Medicine, University of Tartu, Ravila 19, 50411 Tartu, Estonia; reet.mandar@ut.ee (R.M.); siiri.koljalg@ut.ee (S.K.); 5Laboratory of Clinical Microbiology, United Laboratories, Tartu University Hospital, L. Puusepa 1a, 50406 Tartu, Estonia

**Keywords:** eucalypt leaves extract, complex processing, phenolics, terpenes, polyethylene oxide, aqueous gel, semi-solid extrusion 3D printing

## Abstract

The pharmaceutical industry usually utilizes either hydrophobic or hydrophilic substances extracted from raw plant materials to prepare a final product. However, the waste products from the plant material still contain biologically active components with the opposite solubility. The aim of this study was to enhance the comprehensive usability of plant materials by developing a new no-waste extraction method for eucalypt leaves and by investigating the phytochemical and pharmacological properties of eucalypt extracts and their 3D-printed dosage forms. The present extraction method enabled us to prepare both hydrophobic soft extracts and hydrophilic (aqueous) dry extracts. We identified a total of 28 terpenes in the hydrophobic soft extract. In the hydrophilic dry extract, a total of 57 substances were identified, and 26 of them were successfully isolated. The eucalypt extracts studied showed significant antimicrobial activity against *Staphylococcus aureus*, *Pseudomonas aeruginosa*, *Bacillus subtilis*, *Candida albicans*, *Corynebacterium diphtheriae gravis*, and *Corynebacterium diphtheriae mitis*. The anti-inflammatory activity of the dry extract was studied using a formalin-induced-edema model in mice. The maximum anti-exudative effect of the dry extract was 61.5% at a dose of 20 mg/kg. Composite gels of polyethylene oxide (PEO) and eucalypt extract were developed, and the key process parameters for semi-solid extrusion (SSE) 3D printing of such gels were verified. The SSE 3D-printed preparations of novel synergistically acting eucalypt extracts could have uses in antimicrobial and anti-inflammatory medicinal applications.

## 1. Introduction

Due to the limited and steadily decreasing supply of natural resources, there is an emergent interest in discovering new medicines through the complex processing of raw plant materials [1,2,3]. This approach allows researchers to increase the range of herbal remedies, to make rational use of plant resources, to increase the profitability of production, and to decrease its negative effects on the environment [1,4].

The pharmaceutical industry, as a rule, obtains only one drug product from a given plant raw material, using either hydrophobic or hydrophilic substances to extract the final product. The remainder of the plant material is thrown away as waste, although it still contains a significant number of biological active substances (BAS) with the opposite solubility [5,6,7]. It is evident, however, that combining these substances in one final drug product can increase the effectiveness of pharmacotherapy.

The genus *Eucalyptus* L’Her (*Myrtaceae*) is characterized by a wide variety of species, almost 500 in total. Eucalypt leaves have been used as medicinal raw materials for preparing hydrophobic BAS containing essential oils, tinctures, and “Chlorophyllipt” soft extracts (GNCLS Research Plant LLC, Kharkiv, Ukraine), which are used as herbal medicines [8]. The state-of-the-art literature reveals that the species of this genus contain mainly terpenes, polyphenolic compounds, flavonoids, stilbenes, polysaccharides, resins and waxes. Monoterpenoids are the dominant components of the essential oils of most of the plants in the *Eucalyptus* genus [9,10,11]. The composition of phenolic compounds in *E. globulus*, *E. viminalis*, *E. incrassata*, *E. blakelyi*, *E. rubida*, and *E. grandis* has been partially clarified [12,13,14,15,16,17,18,19,20,21].

The main therapeutic activity of eucalypt medicinal preparations is antiseptic. However, medicinal products containing eucalypt have also a pronounced anti-inflammatory effect [21,22]. Furthermore, these preparations have shown to increase tonic activity, promote rapid healing of wounds, and act as analgesics, weak sedatives, and mild expectorants. In addition, eucalypt medicinal products are widely used for the treatment of infectious and inflammatory diseases of the upper respiratory tract, as a component of complex therapy for influenza, and in the treatment of SARS, neuralgia, myalgia, arthritis, rheumatism, sciatica, sports injuries, infections caused by *Staphylococcus aureus*, etc. [21,22,23].

The Ukrainian pharmaceutical industry produces the anti-staphylococcal plant medicine “Chlorophyllipt” in various dosage forms, such as 1% ethanolic and 2% oil solutions, 0.25% solution for injections, spray, tablets and suppositories [8]. The main active substances in these medicinal products are chlorophylls *a*, *b* and terpenes. The extraction of active substances from eucalypt leaves is carried out with 96% ethanol by using a six-fold re-percolation method with an 18 h infusion at each stage. In the modification stage of extraction, a 4% copper sulphate solution and benzene are used [24]. Annually, 25–30 tonnes of eucalypt leaves are imported to Ukraine for the production of “Chlorophyllipt”; the residual meal becomes waste, although it still contains a significant amount of hydrophilic BAS. Thus, only isoprenoid compounds can be extracted in this six-fold re-percolation process. Therefore, there is an urgent need to study the BAS of eucalypt leaves in order to develop potential new medicines with antimicrobial and anti-inflammatory activity. Combining the hydrophobic and hydrophilic BAS of eucalypt leaves through complex processing enables us to formulate appropriate dosage forms and thus to accomplish our goal.

Three-dimensional (3D) printing is a promising new approach for the on-the-fly formulation of plant extracts, as it enables the preparation of individualized dosage forms with high accuracy and flexibility in the composition of the final dosage form. Such 3D-printed plant products are also highly innovative and sophisticated. To date, synthetic drug substances have been mainly used as an active agent in 3D-printed drug products, and there is little knowledge about the 3D printing of plant extracts and substances. Plant-origin materials are very often sensitive to organic solvents and high temperatures, which creates an additional challenge for a 3D-printing process. Only a few research papers on the pharmaceutical 3D printing of plant extracts have been published to date [25,26,27]. A semi-solid extrusion (SSE) 3D-printing method is the most suitable for plant-origin materials because it can be performed at room temperature without any heating [28,29,30].

The aim of this study was to promote the usability of plant materials in a pharmaceutical formulation by developing a new no-waste extraction method for eucalypt leaves for preparing both hydrophobic soft eucalypt extracts and hydrophilic dry extracts. Moreover, the phytochemical and pharmacological properties of such extracts and their SSE 3D-printed dosage forms were investigated. The antimicrobial and anti-inflammatory activities of extracts and 3D-printed preparations were studied.

## 2. Results

The hydrophobic soft eucalypt extract was prepared using the “Chlorophyllipt” approach [24]. The extract was characterized as a semi-solid viscous mass with a dark green color and a specific smell.

The hydrophilic dry eucalypt extract was prepared from the waste raw materials that were obtained after the extraction of the above-mentioned soft extract. The dry extract was a hygroscopic powder with a color ranging from light-brown to brown and a specific smell. The color depended on the quality of raw materials and drying conditions. The yield of hydrophilic dry extract was 9%.

### 2.1. Phytochemical Study of the Eucalypt Extracts

The main terpenoids of hydrophobic soft eucalypt extract were quantitatively determined by means of a gas chromatograp (GC) equipped with a mass-spectrometric (MS) detector. The results are presented in Table 1.

A total of 28 substances were identified in the hydrophobic soft eucalypt extract. The quantitative content of chlorophyll derivatives in the soft extract was 3.31 ± 0.01%. The content of terpenes was 4.3 ± 0.02%, and essential oil (by hydro-distillation) 3.2 ± 0.1%.

Figure 1 shows the approach used for the isolation and identification of the main components of the hydrophilic (aqueous) dry extract of eucalypt leaves. All substances that were isolated and identified in the dry eucalypt extract are listed in Table 2. A total of 57 substances were identified in the hydrophilic (aqueous) dry eucalypt extract, and 26 of them were isolated individually. From the dry extract of eucalypt leaves, two phenolic acids (gallic and ellagic acid), five hydroxycinnamic acids (*p*-coumaric, ferulic, caffeic, chlorogenic, and neochlorogenic acid), six coumarins (coumarin, umbelliferon, scopoletin, daphnoretin, skymin and scopoline), eight flavonoids (luteolin, kaempferol, myricetin, quercetin, isoquercitrin, astragalin, isorhamnetin, and isorhamnetin 3-*O*-β-d-glucopyranoside), two triterpenoids (ursolic and oleanolic acids), two organic acids (tartaric and malic acid), and one urea derivative (allantoin) were isolated. In addition, four sugars and 22 amino acids were identified.

These results allowed us to select and adapt methods by which to identify the polysaccharides, amino acids, flavonoids and tannins in the dry extract of eucalypt leaves and also enabled us to select standards for quantification. The BAS groups and individual substances found in the dry eucalypt extract were shown to have antimicrobial (tannins, flavonoids) and anti-inflammatory activity (polyphenolic compounds, polysaccharides).

For further standardization of the dry eucalypt extract, the amino acids, polysaccharides, hydroxycinnamic acids, flavonoids and polyphenolic compounds in the extract were identified (Table 3).

In our previous study, we developed a SSE 3D-printable PEO-eucalypt-extract gel based on a 20% aqueous PEO solution [31,32]. In this study, the hydrophilic dry eucalypt extract, which has pronounced anti-inflammatory activity, was added to this gel composition at a concentration of 10 mg/mL. The physical appearance and homogeneity of aqueous PEO-eucalypt extract gels were investigated by visual inspection and light microscopy. As shown in Figure 2, the aqueous PEO-eucalypt-extract gels based on 20% PEO solution were homogeneous and uniform in structure.

The PEO gels loaded with both hydrophobic soft eucalypt extract and hydrophilic dry extract were successfully 3D printed to composite small discs (10 mm diameter). The discs were used to study the antimicrobial activity of these dosage forms.

### 2.2. Pharmacological Studies of the Eucalypt Extracts

#### 2.2.1. Study of Antimicrobial Activity

A total of 13 standard (culture collection) strains of microorganisms and 8 clinical strains obtained from patients with such conditions as tonsillitis, bronchitis, purulent wounds, burns, vaginitis, and pneumonia were used in the experiments. Six trials were conducted (n = 6). The dry eucalypt extract was dissolved in water. The test concentrations were 1% and 10%.

The results of the antimicrobial-activity study based on a serial-dilutions method and a diffusion-in-agar method are presented in Table 4 and Table 5.

The eucalypt extracts showed antimicrobial activity against various taxonomic groups of microorganisms. The hydrophilic dry eucalypt extract has significant antimicrobial activity against *S. aureus*, *B. subtilis*, *P. aeruginosa*, *C. diphtheriae* gravis, *C. diphtheriae mitis*, and *C. albicans*.

The SSE 3D-printed round discs loaded with eucalypt extracts showed different growth-inhibition effects on the *S. aureus* and *S. pyogenes* strains used in an agar-diffusion test (the maximum growth-free zones around the substance were 4 mm and 3 mm, respectively). As seen in Table 6, a similar (but weaker) effect was observed with *S. mutans* and *S. sobrinus* strains (with growth-inhibition zones up to 1 mm). The eucalypt-extract discs did not have any effect on the growth of *E. coli* and *C. albicans*. The two most effective eucalypt-extract preparations were the one that contained 10 mg/mL of hydrophobic soft eucalypt extract and the one with 10 mg/mL each of both eucalypt extracts in 20% PEO (Table 6).

#### 2.2.2. Anti-Inflammatory Activity of Dry Eucalypt Extract

The anti-inflammatory activity of the hydrophilic dry eucalypt extract was studied with a formalin-induced-edema model in white mice (17–22 g). The results are shown in Table 7.

The results showed that the hydrophilic dry eucalypt extract had pronounced anti-inflammatory activity in mice. The maximum value for an antiexudative effect (61.5%) was found after the hydrophilic dry eucalypt extract was administered to mice at a dose of 20 mg/kg. Perhaps surprisingly, the level of anti-inflammatory activity of the dry eucalypt extract was comparable to the activity of a reference drug (diclofenac sodium, Voltaren^®^) at the doses studied.

## 3. Discussion

The present novel binary no-waste extraction method for eucalypt leaves enabled us to obtain two extracts, one containing hydrophobic substances (a soft extract), and the other containing hydrophilic substances (a dry extract). The proposed method is simple and can be readily implemented in pharmaceutical-industry plants (e.g., in Ukraine) using standard equipment. Purified water was used as an extractant, thus making the process cheap and environmentally safe because it does not require working with poisonous, flammable, and harmful solvents and reagents. The extraction method utilizes (as a raw material) the parts of eucalypt leaves that remain after the hydrophobic fraction has been extracted. The implementation of binary extraction such as that developed for this plant material results in more rational use of natural resources, increases the profitability of production and reduces the negative impact on the environment. Moreover, the extracts prepared by the present no-waste method are most likely applicable as a BAS-rich material for the preparation of various medicinal and food-supplement products.

In the present study, we found that the main BASs in the hydrophobic soft extract are terpenes and chlorophylls, which have been reported to have significant antimicrobial activity [22,33,34]. Our results indicated that such BASs have significant antimicrobial activity against *S. aureus.*

The following BASs were isolated from the hydrophilic dry extract of *Eucalyptus viminalis* Labill. leaves for the first time ever: two phenolic acids (gallic and ellagic acid), five hydroxycinnamic acids (*p*-coumaric, ferulic, caffeic, chlorogenic, and neochlorogenic acid), six coumarins (coumarin, umbelliferon, daphnoretin, scopoletin, skymin and scopolin), eight flavonoids (luteolin, astragalin, kaempferol, myricetin, quercetin, isoquercitrin, isorhamnetin, and isorhamnetin 3-*O*-β-d-glucopyranoside), two triterpenoids (ursolic and oleanolic acids), two organic acids (tartaric and malic acid), and one urea derivative (allantoin). In addition, four sugars and 22 amino acids were identified. The BASs found in the hydrophilic eucalypt extract have been shown to possess antimicrobial and anti-inflammatory activity [3,4,21]. In the present study, we confirmed experimentally that such BASs have the above-mentioned activities.

The hydrophilic dry eucalypt extract showed a broader range of antimicrobial activity compared to the hydrophobic soft extract. The hydrophilic dry extract (unlike the hydrophobic soft extract) had clear antimicrobial activity against, e.g., *B. subtilis*, *P. aeruginosa*, *C. diphtheriae gravis*, *C. diphtheriae mitis*, and *C. albicans*. Moreover, the hydrophilic dry extract had pronounced anti-inflammatory activity. The museum strains, compared to the clinical ones, were more sensitive to the eucalypt extracts. For example, the eucalypt extracts inhibited the growth of museum strains of *S. aureus* at a concentration of 25–35 mg/mL, but inhibited the growth of clinical strains at 50–70 mg/mL; similar results were found for *E. coli*—35–45 mg/mL and 60–80 mg/mL, respectively, and for *P. aeruginosa*—50–55 mg/mL and 80–150 mg/mL, respectively. This difference may be explained by the greater resistance of clinical strains to antimicrobial agents. These results indicate the feasibility of combining these extracts in one dosage form. Such a medicinal product could be used to, e.g., enhance wound healing and relieve the inflammation of damaged tissue.

In this study, we used SSE 3D printing technology to prepare novel eucalypt-extract-loaded dosage forms for potential medicinal applications. Previously, we found the optimal printing parameters and the optimized composition of a basic PEO gel loaded with a hydrophobic soft eucalypt extract for SSE 3D printing [31,32]. Based on the previous formulation, we were able to successfully 3D-print the novel types of round scaffolds (small discs) loaded with the hydrophilic dry eucalypt extract at different concentrations. In addition, the 3D-printed composite scaffolds (small discs) loaded with both hydrophilic dry extract and hydrophobic soft extract were prepared and their levels of antimicrobial activity were evaluated in vitro.

The antimicrobial effect of the 3D-printed preparations of eucalypt extracts was found to be specific to the microbial species, with the extracts being more effective against Gram-positive cocci. Eucalypt extracts inhibited the growth of the main pathogens that cause wound and upper-respiratory-tract infections, such as *S. aureus* and *S. pyogenes*. The extracts also showed inhibiting activity against caries-causing pathogens, such as *S. mutans* and *S. sobrinus*. Similar activity against *S. aureus* was observed in our previous study [32]. According to the literature, eucalypt extracts possesses some capacity for growth inhibition against different streptococci [33,35]. However,, at this time, virtually nothing is known about the effect of 3D-printed preparations (scaffolds) loaded with eucalypt extracts on the viability of *S. pyogenes*, *S. mutans*, and *S. sobrinus*.

In this study, we used both reference strains and clinical strains of several bacteria and one yeast in order to investigate the antimicrobial effect of eucalypt extracts. It is well known that the selected set of target microorganisms can cause several infections of the oral cavity and upper respiratory tract. For example, *S. pyogenes* is associated with tonsillitis, and *S. mutans* and *S. sobrinus* are associated with dental caries. *Candida albicans* causes candidiasis of the oral cavity and other locations, and *C. diphtheriae* is the causative agent of diphtheria. The microorganisms used in our study can also cause intestinal infections (*S. enterica*, *S. flexneri*), wound infections, and other opportunistic infections (*S. aureus*, *P. vulgaris*, *K. pneumoniae*, *E. coli*, *P. aeruginosa*, *B. subtilis*). In this study, we found that the 3D-printed discs (scaffolds) loaded with a hydrophilic dry eucalypt extract showed weaker antimicrobial activity compared to the corresponding 3D-printed preparations loaded with a hydrophobic soft extract. However, as the hydrophilic dry extract has significant anti-inflammatory activity, it would be justified to combine these two extracts in one 3D-printed preparation to achieve a synergistic effect.

## 4. Materials and Methods

### 4.1. Raw Materials

*Eucalyptus viminalis* Labill. leaves (Ltd “LIKTRAVY”, Zhytomyr, Ukraine) were used as a raw material for preparing the plant extracts. Their use complies with the requirements of the State Pharmacopeia of Ukraine [36].

### 4.2. Preparation of Eucalypt Extracts

The hydrophobic soft and hydrophilic (aqueous) dry extracts of eucalypt leaves were prepared according to the process flow shown in Figure 3.

First, 7 kg of dry eucalypt leaves was crushed by rolling to particle sizes of 2.5–3 mm, and then the crushed leaves were mixed with 21 L of 96% ethanol. A total of 82 L of eucalypt ethanolic extract was obtained after a 4-fold extraction. From this ethanolic extract, “Chlorophyllipt” soft extract was subsequently obtained [24]. After the ethanol was regenerated from the powdered raw material, the material was extracted with 21 L of purified water at 95 °C for 2 h and infused for another 10–12 h. The extraction was repeated thrice [37]. The combined extract (60.5 L) was evaporated at 85 °C in a vacuum circulation apparatus at a pressure of 92 kPa to a water-to-residue ratio of 20:l. This residue was a semi-solid, transparent, dark brown liquid and was kept for 4 days at 4 ± 2 °C. The separated supernatant was sterilized and dried in a spray dryer (EVZ-01-RC-1,2-09-NK-21, “50th anniversary of October” factory, 1989, USSR) with coolant temperatures of 95 °C at the inlet and 65 °C at the outlet to the dry extract.

### 4.3. Preparation of Eucalypt-Extracts-Loaded Gels for 3D Printing

Water gels made with polyethylene oxide (PEO) (MW approx. 900,000, Sigma-Aldrich, Burlington, MA, USA) at a concentration of 20% were used for the SSE 3D printing of the eucalypt extracts. For this purpose, PEO (2.0 g) was dissolved in purified water (5 mL) for at least 13–15 h at a temperature of 20 ± 2 °C to create a viscous gel [31,32,38]. Eumulgin SMO 20 (Polysorbate 80) was used to improve the release of the eucalypt extracts from the 3D-printed scaffolds [31,32]. The soft eucalypt extract (100.0 mg) was solved in 1 mL of ethanol, mixed carefully with Eumulgin SMO 20, and added to 2 mL of water. The dry eucalypt extract (100.0 mg) was dissolved in 2 mL of purified water. The solution containing the soft extract was added to the solution containing the dry extract, and the resulting solution was mixed carefully. Finally, this mixture was added to the PEO gels.

### 4.4. 3D Printing of the Eucalypt Extracts

The PEO gels with the eucalypt extracts were printed using a bench-top SSE 3D printing system (System 30 M, Hyrel 3D, Norcross, GA, USA). The printing head includes a steel syringe. Its plunger is united to a stepper motor. A blunt needle (Gauge, 21G) fixed to a syringe is used as a printing nozzle. The printing head was not heated. The printing-plate temperature was 30 °C. A printing head moved at a set speed of 0.5 mm/s on the X–Y axis during SSE 3D printing. The software of an SSE 3D printer regulates the temperature of the plate, the gel-extrusion rate, and the printing-head speed. A total of five (5) layers were printed in preparing each round disc, and the present 3D-printed discs were used in a microbiological study. A round scaffold (diameter 10 mm) was created using FreeCAD software (vers. 0.19/release date 2021) [39]. The 3D-printed PEO scaffolds were weighed using an analytical scale (Scaltec SBC 33, Scaltec, Burladingen, Germany).

### 4.5. Phytochemical Analysis

Three different plates, “Silufol UV-254”, “Silufol UV-366” and “Sorbfil”—PTSH-A-UV, were used for thin-layer chromatography (TLC). The ascending and descending one-dimensional, two-dimensional, and multiple TLC methods were used. The *R*_F_ values on the chromatograms are the average values of 5–6 measurements.

The solvents used in TLC were applied in volume units. The following solvent systems were used for TLC: # 1, *n*-butanol–acetic acid–water (4:1:2); # 2, 30% acetic acid; # 3, 15% acetic acid; # 4, hexane (formamide 25%); # 5, chloroform (formamide 25%); # 6, toluene–*n*-butanol (3:1)/water (35%); # 7, chloroform–acetic acid–water (13:6:2); # 8, toluene–ethyl acetate–acetic acid (12:4:0.5); # 9, ethyl acetate–formic acid–water (3:1:1).

Cellulose (0.25–0.5 mm fraction), KSK-brand silica gel with a particle size of 0.25 to 0.75 mm, LS 100/250 brand, and polyamide sorbent powder with particles 0.25–0.75 mm were used for column chromatography.

The substances were analyzed after two- and three-fold crystallization from the appropriate solvents and after drying in a vacuum at 1.3–1.6 kPa over phosphoric anhydride for 4 h at 110–115 °C. The melting point was determined by a capillary method [36]. UV-absorption spectra and the optical density of solutions were recorded with a Hewlett Packard 8453 spectrophotometer (Santa Clara, CA, USA), using cuvettes with a layer thickness of 10 mm. IR spectra were recorded with a UR-20 spectrometer (Carl Zeiss, Jena, Germany) using potassium bromide tablets (1 mm high) at a ratio of substance to filler of 1:200–1:400. The optical activity of the glycosides was measured with a SPU-E polarimeter (Schmidt+Haensch GmbH & Co., Berlin, Germany).

*Qualitative and quantitative analysis of terpenoids.* The terpenoids of the soft eucalypt extract were quantitatively measured by means of a gas chromatograph (Agilent Technology 6890, Santa Clara, CA, USA) equipped with a 5973 mass-spectrometric (MS) detector. The following standard conditions were used: HP-5 silica capillary columns with bonded stationary phases SPB-5 (30 m × 0.25 mm, 0.25 μm, Supelco, Bellefonte, PA, USA). The temperature was increased from 50 °C to 250 °C at 4 °C/min, with the injector temperature fixed at 250 °C. The injection volume was 2.0 μL, injected twice. He was used as a carrier gas (split ratio 150:1) at a flow rate of 1 mL/min. The temperature of the interface was 250 °C, and that of the ion source was 230 °C. In the MS detector, 70 eV electrons were used for ionization at a scan rate of 2 scans/s, with an acquisition mass range of 29–450 a.m.u. Compounds were identified by comparing the obtained mass spectra with the ones from the NIST05-WILEY library (approximately 500,000 mass spectra) [40]. The components’ retention indices were determined based on the results of control analyses with normal alkanes (C10–C18).

*Qualitative and quantitative analysis of free amino acids* in the dry eucalypt extract was conducted using an amino-acid analyzer T 339 (Mikrotekhnika, Prague, Czech Republic) [41]. In the analysis, we used columns with a water jacket (0.37 × 25 cm) and ionite osteon LGANB. A sequence of lithium-citrate buffer solutions of different acidities and ionic strengths (pH 2.9 + 0.01; 2.95 + 0.01; 3.2 + 0.02; 3.8 + 0.02 and 5 + 0.02) was used as a mobile phase. The flow rate of the buffer solutions was 12 mL/h, and the reagent rate was 8–10 mL/h. Detection was carried out using post-column staining with a solution of ninhydrin in DMSO at a wavelength of 570 nm. Quantification was performed using standard amino-acid solutions (TU 6-09-3147-83).

*Study of the quantitative content of amino acids by a spectrophotometric method.* A spectrophotometric method was applied to quantify the amino acids in the dry eucalypt extract. Because the average molecular weight of amino acids in eucalypt leaf extract is 137 g/mol, which is closest to the value of the molecular weight of leucine (131 g/mol), the quantitative analysis was carried out based on this amino acid [42].

*Quantification of polysaccharides by a gravimetric method*. Approximately 1.0 g (exact weight) of the extract was added to 10 mL of water. The flask was then connected to a reflux condenser and boiled with stirring for 30 min. Polysaccharides were precipitated three times using ethanol, and the precipitate was separated by centrifugation at a rotation speed of 5000 rpm for 10 min. Finally, the precipitate was dried to a constant mass and weighed on an analytical balance [36].

*Quantification of polysaccharides by a spectrophotometric method*. Determination of polysaccharide content in the extract of eucalypt leaves was carried out by the spectrophotometric method in terms of glucose units after sulfuric-acid hydrolysis for 2.5 h, followed by neutralization and reaction with a 1% aqueous solution of picric acid [43].

*Quantification of hydroxycinnamic acids.* Caffeic, *p*-coumaric, ferulic, neochlorogenic and chlorogenic acids were found in the eucalypt-leaf extract by a TLC method. The largest area of spots represented the highest content of chlorogenic acid in TLC. The content of hydroxycinnamic acids in the dry extract of eucalypt leaves was established by a spectrophotometric method using chlorogenic acid as a standard. The absorption maximum of the chlorogenic acid standard was observed at 327 nm, so the measurements were performed at this wavelength [3,42,44].

*Quantification of flavonoids.* Previous studies have shown the presence of a total of eight (8) flavonoid compounds (one flavone and seven flavonols, Table 1) in the dry extract. To verify the correlation of the results from different methods and the validity of the standard used (s well as conversion validity), the content of flavonoids was determined by a spectrophotometric method based on rutin and quercetin [42,45,46,47].

*Determination of tannin content* by a complexometric method. A complexonometric method was selected for the quantification of tannins because it is more selective for tannins in a mixture with other substances (flavonoids, phenolic acids, etc.) [48,49,50].

*Determination of the total content of polyphenolic compounds* by a spectrophotometric method based on gallic acid. This method was selected because gallic acid is the main component of polyphenolic compounds [42,44,51].

*Quantification of chlorophylls.* Chlorophyll content was determined by a spectrophotometric method at wavelengths of 649 and 665 nm [31,32,34].

### 4.6. Study of Antimicrobial Activity

The research on the antimicrobial activity of the eucalypt extracts was conducted using serial dilutions in a liquid nutrient medium and diffusion in agar at the I. Mechnikov Institute of Microbiology and Immunology (Kharkov, Ukraine) under the supervision of Ph.D. Osolodchenko T.P. [52,53,54].

In accordance with the WHO recommendations, the reference strains *Staphylococcus aureus* ATCC 6538, *Staphylococcus aureus* ATCC 25923, *Escherichia coli* ATCC 25922, *Pseudomonas aeruginosa* ATCC 27853, *Pseudomonas aeruginosa* ATCC 9027, *Proteus vulgaris* NSTC 4636, *Bacillus subtilis* ATCC 6633, and *Candida albicans* ATCC 885/653, were used to estimate the antimicrobial activity of the extracts. In addition, the study was conducted on clinical strains that were already in the laboratory: *Shigella flexneri* 170, *Salmonella enterica* Typhimurium 144, *Salmonella enterica* Paratyphi A-290, *Corynebacterium diphtheriae gravis* 14 tox+, *C. diphtheriae mitis* 6 tox+. Nutrient broth was used for the cultivation of microorganisms, and glucose was added at a concentration of 3 mL per 100 mL of broth for cultivation.

*The method of serial dilutions* enabled the quantification of antimicrobial activity for each of the extracts. The meat-peptone nutrient medium was poured into 10 test tubes of 2 mL each. A total of 2 mL of the prepared eucalypt extract solution was added to the first test tube and mixed, and 2 mL was taken and transferred to the next test tube, etc. Then, 2 mL of the mixture was drained from the last test tube. Next, 0.2 mL of the prepared culture was added to each test tube (including the control one) at a concentration of 10^7^–10^8^ CFU/mL. The standard of the seed crop was established according to the optical standard of turbidity of DIX, which was named after L. O. Tarasevich. The seed crops were placed in a thermostat for 18–24 h at 37 °C. The results were determined by the presence or absence of turbidity in the medium in test tubes containing different dilutions of the experimental extract. The concentration of the drug in the last test tube with a transparent medium (with transparency indicating absence of growth of the test microbe) corresponds to the minimum inhibitory concentration (MIC) of the drug.

To determine the bactericidal concentration, 0.1 mL of a transparent medium from the last two to three tubes (with transparency indicating an absence of visible growth), was placed on the cups with a dense nutrient medium or tubes with broth. Incubation was carried out for 18–24 h at 37 °C, and the minimum concentration of the drug was recorded (based on the sample that resulted in no growth on agar or in broth). The amount of drug present in that sample corresponds to the drug’s minimum bactericidal concentration.

*Extracts diffusion in agar* was conducted using “wells”. Determination of the extracts’ activity was carried out on two layers of a dense nutrient medium poured into a Petri dish. “Hungry” non-seeded media (agar-agar, water, salts) were used in the lower layer. The bottom layer was a substrate (height, 10 mm) on which 3–6 thin-walled stainless steel cylinders (8 mm diameter; 10 mm height) are installed strictly horizontally. The cylinders were covered with an upper layer consisting of nutrient agar medium and then melted and cooled to 40 °C. In this medium, the appropriate standard daily culture of the test microbe was placed. Previously, the upper layer had been homogenized. After hardening, the cylinders were pulled out of the hole that had formed with sterile tweezers, and the tested substance was introduced at the appropriate volume. The thickness of the medium for the upper layer ranged from 14 to 16 mm. The cups were dried for 30–40 min at ambient room temperature and placed in a thermostat for 18–24 h.

To investigate new antibacterial agents and antibiotic-resistant strains, the following criteria were used: the absence of a growth-inhibition zone around the hole or a growth-inhibition zone up to 10 mm in diameter indicate that the microorganism is not sensitive to the drug or is not sensitive to the drug at that concentration; a growth-inhibition zone with a diameter of 10–15 mm indicates that the culture has low sensitivity to the tested concentration of the antibacterial substance; a growth-inhibition zone with a diameter of 15–25 mm indicates that the microorganism is sensitive to the tested medicinal product; a growth-inhibition zone with a diameter exceeding 25 mm indicates high sensitivity of the microorganism to the experimental drug.


*The method for investigating antimicrobial activity of the 3D-printed forms.*


The antimicrobial effect of eucalypt extract was investigated against a total of six (6) microbial species, each represented by three culture collection strains. The microbial species used were *Staphylococcus aureus* (ATCC 29213, HUMB 19417, HUMB 5630); *Escherichia coli* (ATCC 25922, HUMB 7024, HUMB 5666); *Candida albicans* (ATCC 10231, HUMB 05355, HUMB 19373); *Streptococcus pyogenes* (DSM 25943, HUMB 18939, HUMB 18966); *Streptococcus mutans* (HUMB 13034, HUMB 13076, HUMB 13033); and *Streptococcus sobrinus* (HUMB 13087, HUMB 13104, HUMB 13038).

A suspension of microbes was prepared in PBS buffer with a density of 0.5 McFarland and applied to the following agar plates: Mueller-Hinton agar (*S. aureus*, *E. coli*, *C. albicans*) and Mueller-Hinton agar with blood (*Streptococcus* spp.). The 3D-printed eucalypt-extract molds were placed on the top of agar and incubated in a 10% CO_2_ incubator for 22 h. The diameter of a growth-free zone around the eucalypt 3D-printed preparation was measured.

### 4.7. Study of Anti-Inflammatory Activity

The pharmacological studies were conducted in compliance with the rules of the “European Convention for the Protection of Vertebrate Animals Used for Experimental and Other Scientific Purposes” (Strasbourg, 1986), Directive 2010/63/EU of the European Parliament, and the Council of the European Union (2010) on the protection of animals used for scientific purposes. The Order of the Ministry of Health of Ukraine No. 944 “On Approval of the Procedure for Preclinical Study of Medicinal Products and Examination of Materials for Preclinical Study of Medicinal Products” (2009) and the Law of Ukraine No. 3447-IV “On the protection of animals from cruel treatment” (2006) were also strictly followed. The present research work was approved by the Bioethics Commission of the National University of Pharmacy (protocol No 4 from 3 October 2023) [55,56,57,58,59].

The anti-inflammatory activity of dry eucalypt extracts was studied with white mice (17–22 g) using a formalin-induced-edema model [52,60]. Diclofenac sodium (Voltaren^®^) was chosen as a reference drug. The mice were divided into three groups (n = 8): a control group, a group treated with the eucalypt extract, and a group treated with the reference drug.

The level of anti-inflammatory activity of the extract was evaluated based on the anti-exudative effect. To reproduce acute aseptic exudative inflammation, a 2% formalin solution was injected (0.05 mL) subplantarly one hour after oral administration of dry eucalypt extract, with diclofenac sodium used as a reference drug and water used in the control group. The anti-inflammatory activity of the experimental agents was estimated by their ability to decrease the development of edema in comparison with the control.

### 4.8. Statistical Analysis

The statistical properties of random variables (with an *n*-dimensional normal distribution) are given by their correlation matrices, which can be calculated from the original matrices. A statistical assessment of the data was performed using MS Excel (Microsoft Excel 2016, version 16.0, Microsoft Corporation, Redmond, DC, USA). *p*-values less than 0.05 were considered statistically significant [36,61].

## 5. Conclusions

The present study introduces a new no-waste extraction method for eucalypt leaves that enables researchers to prepare both hydrophobic soft extracts and hydrophilic dry extracts with potential pharmacological activity. Such an extraction method could support the pharmaceutical industry in using plant materials for formulation development in a more comprehensive way. A total of 28 terpenes were identified in the hydrophobic soft extract. In the hydrophilic dry extract, a total of 57 substances were identified, and 26 of them were isolated individually. The eucalypt extracts and their 3D-printed dosage forms have antimicrobial and anti-inflammatory activity. The eucalypt extracts (especially the hydrophobic soft extract) present antimicrobial activity against various taxonomic groups of microorganisms such as *S. aureus*, *B. subtilis*, *P. aeruginosa*, *C. diphtheriae gravis*, *C. diphtheriae mitis*, and *C. albicans*. The hydrophilic dry extract has pronounced anti-inflammatory activity. The maximum anti-exudative effect of the hydrophilic dry extract (61.5%) was observed in mice at a dose of 20 mg/kg. Composite gels of polyethylene oxide (PEO) and eucalypt extract were developed, and key process parameters for the semi-solid extrusion (SSE) 3D printing of such gels were verified. The SSE 3D-printed preparations of synergistically acting eucalypt extracts could have uses in antimicrobial and anti-inflammatory medicinal applications.

## Figures and Tables

**Figure 1 plants-13-00754-f001:**
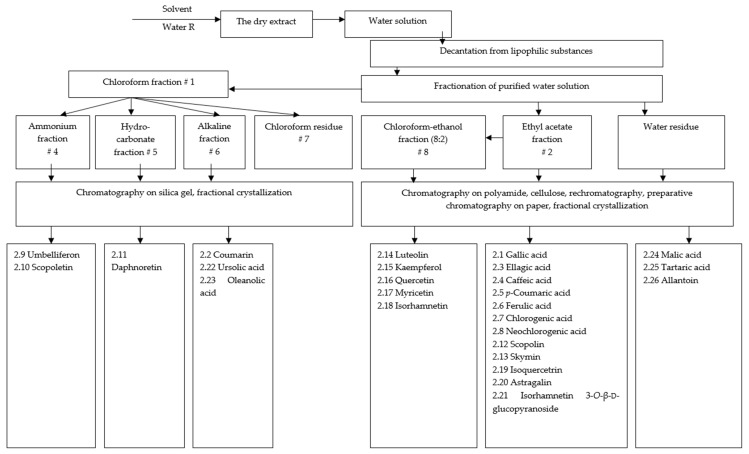
Overview of the process for the isolation and identification of biologically active substances (BAS) from the dry eucalypt-leaf extracts.

**Figure 2 plants-13-00754-f002:**
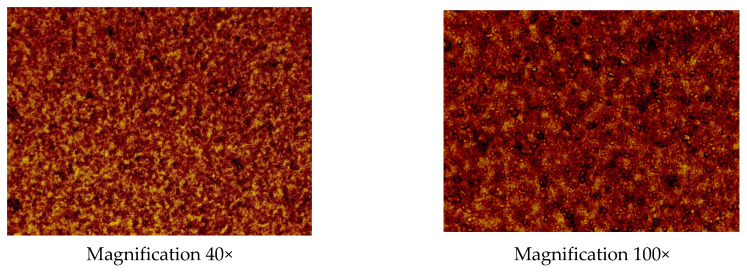
Microscopic analysis of the 20% aqueous polyethylene oxide (PEO) gels loaded with the two eucalypt extracts. Magnifications: 40×, 100×, 400×, and 500×.

**Figure 3 plants-13-00754-f003:**
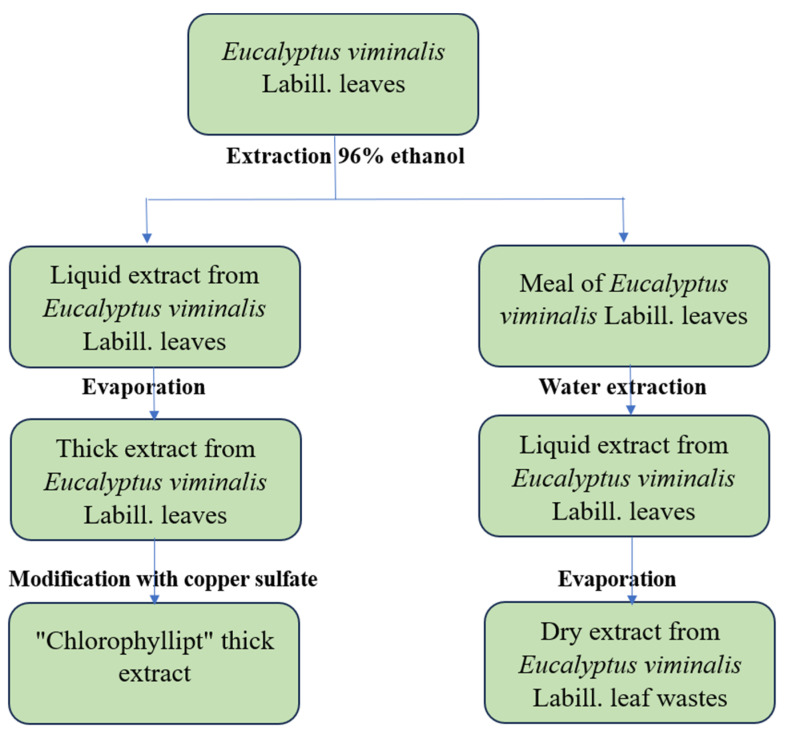
The no-waste extraction process for eucalypt leaves.

**Table 1 plants-13-00754-t001:** Terpenoids of the hydrophobic soft eucalypt extract.

Substance	Assay (%) of the Volatile Fraction of the Extract
α-Phellandrene	3.13
1,8-Cineol	11.15
trans-Pinocarveol	1.48
Pinocarvone	0.41
Terpinen-4-ol	0.68
α-Terpineol	1.5
α-Terpinyl acetate	0.57
Geranyl acetate	0.47
α-Guryunen	1.81
Kalaren	1.15
Aromadendren	26.01
Allo-aromadendrene	3.95
Leden	2.55
Dehydroaromadendrene	0.47
Epiglobulol	3.47
Globulol	14.66
Viridiflorol	2.57
Epi-γ-eudesmol	1.16
Epi-β-eudesmol	1.17
Kubenol	3.29
β-Eudesmol	1,05
α-Eudesmol	1.15
Posifoliol	2.06
Palmitic acid	0.89
Ethyl palmitate	1.01
Phytol	0.51
Ethyl oleate	0.32
Ethyl linolenate	0.68
7 unidentified compounds	4.24

**Table 2 plants-13-00754-t002:** The main physicochemical properties of substances isolated from the dry extract of eucalyptus leaves.

#	Substance	*T*_melt_, °C	${\left[\alpha \right]}_{D}^{20}$, Degree	UV Spectrum, *λ*_max_, nm	*R*_F_ in Solvent Systems	
					System	*R_F_*
Derivatives of benzoic acid						
1.	2.1 Gallic acid (3,4,5-trihydroxybenzoic acid)	226–228	-	272	1	0.65
2.	2.3 Ellagic acid (hexahydroxydiphenic acid dilactone)	360 distr.	-	366	1 2	0.12 0.06
Derivatives of cinnamic acid						
3.	2.4 Caffeic acid (3,4-dihydroxycinnamic acid)	194–195	-	325 300 235	1 3	0.8 0.5
4.	2.5 p-Coumaric acid (4- hydroxycinnamic acid)	212–214	-	310 228 217	1 3	0.9 0.6
5.	2.6 Ferulic acid (4-hydroxy-5-methoxy-cinnamic acid)	168–170	-	320 290 234	1 3	0.88 0.55
6.	2.7 Chlorogenic acid (5-*O*-caffeyl-d-quinic acid)	203–205	−32 (methanol)	325 300 245	1 3	0.62 0.7
7.	2.8 Neochlorogenic acid (3-*O*-caffeyl-d-quinic acid)	Amorphous	+2.6 (ethanol)	325 300 243	1 3	0.64 0.75
Coumarin derivatives						
8.	2.2 Coumarin	67–69	-	-	4	0.2
9.	2.9 Umbelliferon (7-hydroxycoumarin)	228–230	-	231 258 327	5	0.36
10.	2.10 Scopoletin (6-methoxy-7-hydroxycoumarin)	202–204	-	230 255 296 346	5	0.58
11.	2.11 Daphnoretin (2-methoxy-6-oxy-3,7′-dicoumarin ether)	254–256	-	-	5	0.85
12.	2.12 Scopolin (6-methoxy-7-(*O*-(-d-glucopyranosyl)-coumarin)	218–220	–8.5 DMF	231 330	6	0.24
13.	2.13 Skymin (7-(*O*-(-d-glucopyranosyl)-coumarin)	218–220	−80 (methanol)	220 252 325	5	0.38
Flavones						
14.	2.14 Luteolin (5,7,3′,4′-tetrahydroxyflavone)	232–241	-	255 318 350 410	1 3	0.82 0.11
Flavonols						
15.	2.15 Kaempferol (3,5,7,4′-tetrahydroxyflavone)	273–274	-	366 266	1 7	0.83 0.55
16.	2.16 Quercetin (3,5,7,3′,4′-pentahydroxyflavone)	310–312	-	375 268 256	1 7	0.69 0.32
17.	2.17 Myricetin (3,5,7,3′,4′,5′-hexahydroxyflavone)	350–354 distr.	-	374 272 254	7	0.18
18.	2.18 Isorhamnetin (3,5,7,4′-tetrahydroxy-3′-methoxyflavone)	167–170	-	370 265 254	1 7	0.85 0.73
Flavonol glycosides						
19.	2.19 Isoquercetrin (quercetin-3-*O*-β-d-glucopyranoside)	227–229	−12.5 (methanol)	355 267 256	1 3	0.52 0.36
20.	2.20 Astragalin (kaempferol-3-*O*-β-d-glucopyranoside)	196–198	−6.8 (ethanol)	375 270	1 3	0.69 0.37
21.	2.21 Isorhamnetin 3-*O*-β-d-glucopyranoside	317–319	−30 (DMF)	357 302 255	1 3	0.46 0.59
Triterpenoids						
22.	2.22 Ursolic acid	280–283	+62.5 (chloroform)	-	1	0.89
23.	2.23 Oleanolic acid	300–303	+79.0 (chloroform)	-	1 8	0.9 0.44
Organic acids						
24.	2.24 Malic acid	100–101				
25.	2.25 Tartaric acid	170–171	+11.9 (ethanol)	-	9	0.48
Monosugars						
26.	D-glucose				1	0.23
27.	D-galactose				1	0.17
28.	D-xylose				1	0.31
29.	L-rhamnose				1	0.43
Amino acids						
30.	Cysteine *					
31.	Taurine *					
32.	Phosphoethanolamine *					
33.	Aspartic acid *					
34.	Threonine *					
35.	Serin *					
36.	Asparagine *					
37.	Glutamic acid *					
38.	Proline *					
39.	Glycine *					
40.	Alanine *					
41.	Citrulline *					
42.	α-amino-n-butyrin *					
43.	Valine *					
44.	Cystine *					
45.	Cystathionine *					
46.	Methionine *					
47.	Isoleucine *					
48.	Tyrosine *					
49.	Phenylalanine *					
50.	β-Alanine *					
51.	Ethanolamine *					
52.	Ornithine *					
53.	Lysine *					
54.	1-methylhistidine *					
55.	3-methylhistidine *					
56.	Arginine *					
Urea derivatives						
57.	2.26 Allantoin	234–235	-	-	1	0.35

Note: *—the substance was identified using an amino-acid analyzer, model T 339 (Mikrotekhnika, Prague, Czech Republic).

**Table 3 plants-13-00754-t003:** Assay of biologically active substances in the dry extract of eucalypt leaves.

The BAS Group That Was Determined and the Method Used	Assay in the Extract, %
Amino acids:	
Amino-acid analyzer	0.21
Spectrophotometric method for leucine	0.19 ± 0.01
Polysaccharides:	
Gravimetric method	17.42 ± 0.68
Spectrophotometric method for glucose	12.91 ± 0.53
Hydroxycinnamic acid	
Spectrophotometric method for chlorogenic acid	3.38 ± 0.22
Flavonoids:	
Spectrophotometric method for rutin	4.69 ± 0.11
Spectrophotometric method for quercetin	3.7 ± 0.09
Polyphenolic compounds:	
Complexometric method	3.92 ± 0.07
Spectrophotometric method for gallic acid	2.0 ± 0.59

**Table 4 plants-13-00754-t004:** Study of the antimicrobial activity of the dry eucalypt extract by the serial-dilutions method in a liquid nutrient medium.

Microorganisms	MIC of Eucalypt Extract, mg/mL
*S. aureus* ATCC 25923	25–35
*S. aureus* ATCC 6538	25–35
*E. coli* ATCC 25922	35–45
*P. vulgaris* NCTC 4636	45–50
*P. aeruginosa* ATCC 27853	50–55
*P. aeruginosa* ATCC9027	50–60
*B. subtilis* ATCC 6633	25–35
*C. albicans* ATCC 885/653	45–65
*S. typhimurium* 144	45–50
*S. paratyphi* A 290	45–50
*S. flexneri* 170	45–50
*C. diphtheriae gravis* 14 tox+	45–50
*C. diphtheriae mitis* 6 tox+	35–45
*S. aureus* (tonsillitis)	50–70
*S. aureus* (bronchitis)	50–70
*S. pyogenes* (bronchitis)	45–55
*E. coli* (purulent wound)	60–80
*P. aeruginosa* (purulent wound)	60–80
*P. aeruginosa* (burn)	100–150
*C. albicans* (vaginitis)	45–60
*K. pneumoniae* (pneumonia)	100–120

**Table 5 plants-13-00754-t005:** Study of the antimicrobial activity of the dry and soft eucalypt extracts by the diffusion method in agar.

Microorganisms	Growth Inhibition Zones Diameter, mm		
	The Dry-Extract Solution		1% Alcohol Solution of the Soft Extract
	1%	10%	
*S. aureus* ATCC 25923	14	16	23
*S. aureus* ATCC 6538	14	14	24
*E. coli* ATCC 25922	14	15	13
*P. vulgaris* NCTC 4636	13	14	growth
*P. aeruginosa* ATCC 27853	17	18	growth
*P. aeruginosa* ATCC 9027	13	14	growth
*B. subtilis* ATCC 6633	20	22	growth
*C. albicans* ATCC 885/653	17	19	growth
*S. enterica* Typhimurium 144	14	14	ns
*S. enterica* Paratyphi A 290	13	15	ns
*S. flexneri* 170	14	15	ns
*C. diphtheriae gravis* 14 tox+	16	19	ns
*C. diphtheriae mitis* 6 tox+	17	20	ns
*S. aureus* (tonsillitis)	12	14	ns
*S. aureus* (bronchitis)	15	19	ns
*S. pyogenes* (bronchitis)	16	21	ns
*E. coli* (purulent wound)	14	14	ns
*P. aeruginosa* (purulent wound)	15	13	ns
*P. aeruginosa* (burn)	Growth	growth	ns
*C. albicans* (vaginitis)	16	17	ns
*K. pneumoniae* (pneumonia)	14	16	ns

Note: “ns”—not studied.

**Table 6 plants-13-00754-t006:** Susceptibility of microbes to antibiotic activity of the 3D-printed round discs with the dry and soft eucalypt extracts.

Microbe		Inhibition Zone Around the Discs (mm) with the Eucalypt Extracts					
		The Soft Extract, mg/mL	The Dry Extract, mg/mL				The Combined Extract, mg/mL
		10	5	10	20	100	10 + 10
*S. aureus*	ATCC 29213	4	0	0	0	1	2
*S. aureus*	HUMB 19417	4	0	0	0	2	4
*S. aureus*	HUMB 5630	4	0	0	0	4	3
*E. coli*	ATCC 25922	0	0	0	0	0	0
*E. coli*	HUMB 7024	0	0	0	0	0	0
*E. coli*	HUMB 5666	0	0	0	0	0	0
*C. albicans*	ATCC 10231	0	0	0	0	0	0
*C. albicans*	HUMB 05355	0	0	0	0	0	0
*C. albicans*	HUMB 19373	0	0	0	0	0	0
*S. pyogenes*	DSM 25943	3	0	0	0	1	2
*S. pyogenes*	HUMB 18939	3	0	0	0	0.5	2
*S. pyogenes*	HUMB 18966	3	0	0	0	1	2
*S. mutans*	HUMB 13076	1	0	0	0	0	0
*S. mutans*	HUMB 13034	1	0	0	0	0	0
*S. mutans*	HUMB 13033	1	0	0	0	0	0
*S. sobrinus*	HUMB 13087	1	0	0	1	0	0
*S. sobrinus*	HUMB 13104	1	0	0	1	0	0
*S. sobrinus*	HUMB 13038	1	0	1	0	0	0

**Table 7 plants-13-00754-t007:** The anti-inflammatory activity of hydrophilic dry eucalypt extract, evaluated as the level of antiexudative effect.

Agent	Dose, mg/kg	Average Value of Swelling (after 3 h)	Antiexudative Effect, %
Eucalypt extract	20	0.378 ± 0.031	61.5
Voltaren	2.3	0.358 ± 0.061	63.5
Control	0	0.982 ± 0.111	0

## Data Availability

Data is contained within the article.

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
