# Peer review of "Eucalypt Extracts Prepared by a No-Waste Method and Their 3D-Printed Dosage Forms Show Antimicrobial and Anti-Inflammatory Activity"

_plants, 2024, doi:10.3390/plants13060754_

Round 1

Reviewer 1 Report

Comments and Suggestions for Authors

The paper titled: “The eucalypt extracts prepared by a non-wasting method and their 3D-printed dosage forms with an antimicrobial and anti-inflammatory activity” introduces a new extraction method for eucalypt leaves to reduce waste and enhance usability. It yielded both hydrophobic and hydrophilic extracts. The extracts showed significant antimicrobial and anti-inflammatory properties. Additionally, 3D-printed dosage forms containing eucalypt extracts are developed for potential medicinal applications.

This is an interesting paper. It is well written, well discussed, the results are presented in a good manner. The introductory story goes very well. 

However, I have some concerns regarding applied methodology. Why did you not use HPLC for identification and quantification of compounds where relevant instead of TLC?

Can you double check the results of antimicrobial activity by serial dilution technique? How did you obtained values eg. 25-35? Have you used standardized microdilution technique as suggested by CLSI? The values are low and antimicrobial activity seems to be good.

Author Response

Response to the Reviewers

Manuscript ID: plants-2897583

Title: The eucalypt extracts prepared by a non-wasting method and their 3D-printed dosage forms with an antimicrobial and anti-inflammatory activity

Dear Reviewer 1,

The authors of the manuscript thank the reviewers for their helpful comments. We have considered all comments when updating the manuscript and have improved our article based on them. Changes made in the manuscript are marked in yellow. Below, we present our point-by-point responses to all reviewers' comments.

Q0. The paper titled: “The eucalypt extracts prepared by a non-wasting method and their 3D-printed dosage forms with an antimicrobial and anti-inflammatory activity” introduces a new extraction method for eucalypt leaves to reduce waste and enhance usability. It yielded both hydrophobic and hydrophilic extracts. The extracts showed significant antimicrobial and anti-inflammatory properties. Additionally, 3D-printed dosage forms containing eucalypt extracts are developed for potential medicinal applications.

This is an interesting paper. It is well written, well discussed, the results are presented in a good manner. The introductory story goes very well.

R0. The authors of the manuscript thank the reviewer very much for the good words.

Q1. However, I have some concerns regarding applied methodology. Why did you not use HPLC for identification and quantification of compounds where relevant instead of TLC?

R1. Thank you very much for your comment. You are absolutely right about using HPLC, it made the experiment quite easier, but, unfortunately, at that moment there was no access to a HPLC chromatograph, so the experiment was planned according to the available possibilities. Another point was that the PhD student has been learning to isolate substances in an individual state and there was hope to isolate and establish the structure of some absolutely new one, which unfortunately did not come true.

Q2. Can you double check the results of antimicrobial activity by serial dilution technique? How did you obtained values eg. 25-35? Have you used standardized microdilution technique as suggested by CLSI? The values are low and antimicrobial activity seems to be good.

R2. Thank you very much for your comment. We checked the results in the Table 4. We changed µg/mL to mg/mL. It was our technical mistake. Terpenes really have a significant antimicrobial activity. To confirm it, for example The Ukrainian pharmaceutical industry produces the anti-staphylococcal plant medicine "Chlorophyllipt", based on an eucalypt extract in various dosage forms, such as 0.25% solution for injections and spray, where the concentration of the eucalypt extract is 0.2%. So, these incomparable with our results. The description of the method used is described in the section 4.6.

Reviewer 2 Report

Comments and Suggestions for Authors

Manuscript evaluation form

Manuscript entitled The eucalypt extracts prepared by a non-wasting method and their 3D-printed dosage forms with an antimicrobial and anti-inflammatory activity authors: Oleh Koshovyi, Mykola Komisarenko, Tatyana Osolodchenko, Andrey Komissarenko, Reet Mändar, Siiri Kõljalg, Jyrki Heinämäki and Ain Raal is original scientific paper suitable for publication in Plants with very minor revision.

In the present study, a comprehensive approach to the utilization of plant material was made according to the principles of green chemistry. In doing so, they applied the latest SSE 3D technologies in order to determine the antibacterial and anti-inflammatory properties of the obtained extracts or compounds. The complete analysis of all compounds was made according to the highest standards following the quality system for the interpretation of the results.

Finally, the manuscript is easy readable and good organized; all topics are discussed in logical sequence, abstract and introduction are well organized, materials and method are technically accurate in the term of methods and procedures, so I put all corrections in manuscript. Because of all this, I believe that this manuscript corresponds to all the principles of Plants journal.

Author Response

Response to the Reviewers

Manuscript ID: plants-2897583

Title: The eucalypt extracts prepared by a non-wasting method and their 3D-printed dosage forms with an antimicrobial and anti-inflammatory activity

Dear Reviewer 2,

The authors of the manuscript thank the reviewers for their helpful comments. We have considered all comments when updating the manuscript and have improved our article based on them. Changes made in the manuscript are marked in yellow. Below, we present our point-by-point responses to all reviewers' comments.

Manuscript entitled The eucalypt extracts prepared by a non-wasting method and their 3D-printed dosage forms with an antimicrobial and anti-inflammatory activity authors: Oleh Koshovyi, Mykola Komisarenko, Tatyana Osolodchenko, Andrey Komissarenko, Reet Mändar, Siiri Kõljalg, Jyrki Heinämäki and Ain Raal is original scientific paper suitable for publication in Plants with very minor revision.

In the present study, a comprehensive approach to the utilization of plant material was made according to the principles of green chemistry. In doing so, they applied the latest SSE 3D technologies in order to determine the antibacterial and anti-inflammatory properties of the obtained extracts or compounds. The complete analysis of all compounds was made according to the highest standards following the quality system for the interpretation of the results. 

Q1. Finally, the manuscript is easy readable and good organized; all topics are discussed in logical sequence, abstract and introduction are well organized, materials and method are technically accurate in the term of methods and procedures, so I put all corrections in manuscript. Because of all this, I believe that this manuscript corresponds to all the principles of Plants journal.

R2. Thank you very much for your comments. We have taken them all into account and corrected them. mm Hg converted to kPa. Unfortunately we can’t find the correct abbreviation of these journals: Azerbaijan Pharmaceutical and Pharmacotherapy Journal Ceska a Slovenska Farmacie, so left them, like they are.

Reviewer 3 Report

Comments and Suggestions for Authors

25 line (Abstract) - add that plant waste has biologically active components

47 line - confused and unclear sentence

56 and 90 line  - "remaining meal" or (plant meal)- replace the word "meal" with a more acceptable one

64 line -  add that “ Chlorophyllipt" is a herbal medicine (Along with that there is reference number 8 which is not complete, please complete it).

67 line - Rephrase the sentence: write that monoterpenoids are mainly the dominant components of the essential oils of most garden plants of the Eucalyptus genus. It is written this way is confused.

Reformulate throughout the text "eucalipt preparation"

118 line - After “Chloro120 phyllipt” technological scheme, put in brackets that the explanation in Figure 4.

121 - why the smell of 1,8 cineol when it is not the dominant component?

19 line in Results - Insert taht means the ATCC collection of strains

20 line - when you already state that they are clinical isolates, state from which patients they were isolated

34 line - Compare whether there are differences in the activity of the extracts against ATCC strains and clinical isolates. Comment a little more on the obtained result.

In the heading of Tables 5 and 6, indicate which extracts were tested

Comments on the Quality of English Language

Some sentences are confused, interrupted.

Author Response

Response to the Reviewers

Manuscript ID: plants-2897583

Title: The eucalypt extracts prepared by a non-wasting method and their 3D-printed dosage forms with an antimicrobial and anti-inflammatory activity

Dear Reviewer 3,

The authors of the manuscript thank the reviewers for their helpful comments. We have considered all comments when updating the manuscript and have improved our article based on them. Changes made in the manuscript are marked in yellow. Below, we present our point-by-point responses to all reviewers' comments.

Q1. 25 line (Abstract) - add that plant waste has biologically active components

R1. Done.

Q2. 47 line - confused and unclear sentence

R2. The sentence is removed.

Q3. 56 and 90 line  - "remaining meal" or (plant meal)- replace the word "meal" with a more acceptable one

R3. Done.

Q4. 64 line -  add that “ Chlorophyllipt" is a herbal medicine (Along with that there is reference number 8 which is not complete, please complete it).

R4. Done.

Q5. 67 line - Rephrase the sentence: write that monoterpenoids are mainly the dominant components of the essential oils of most garden plants of the Eucalyptus genus. It is written this way is confused.

R5. Done.

Q6. Reformulate throughout the text "eucalipt preparation"

R6. Reformulated.

Q7. 118 line - After “Chloro120 phyllipt” technological scheme, put in brackets that the explanation in Figure 4.

R 7. We added the reference here.

Q8. 121 - why the smell of 1,8 cineol when it is not the dominant component?

R9. 1,8 cineol has the third concentration. The highst one is for aromadendren, but it does not have a strong smell, it is a bit like the smell of mold. Therefore, the smell is provided by 1,8-cineole and globulol. When characterizing eucalypt essential oil, it is customary to write 1,8 cineol. But we corrected it by indicating a specific smell

Q9. 19 line in Results - Insert taht means the ATCC collection of strains

R9. ATCC is a part of the cipher of the museum strain of the microorganism in the catalog, it cannot be deciphered.

Q10. 20 line - when you already state that they are clinical isolates, state from which patients they were isolated

R10. Tables 4 and 5 list diseases for which clinical strains were isolated. As well this information was added into the text.

Q11. 34 line - Compare whether there are differences in the activity of the extracts against ATCC strains and clinical isolates. Comment a little more on the obtained result.

R11. We added to the discusion part such sentances: “The museum strains were more sensitive to the eucalypt extracts compared to the clinical ones. For example, the eucalypt extracts inhibited the growth of S. aureus for the museum strains at a concentration of 25-35 mg/mL, but for the clinical ones - at 50-70 mg/mL; as well for E. coli – 35-45 mg/mL and 60-80 mg/mL, respectively; for P. aeruginosa – 50-55 mg/mL and 80-150 mg/mL, respectively. This may be explained by greater resistance of clinical strains to antimicrobial agents.”

Q12. In the heading of Tables 5 and 6, indicate which extracts were tested.

R12. Done.
